# Experience and Perceptions among Older Outpatients after Myocardial Infarction following an Exercise Intervention: A Qualitative Analysis from the PIpELINe Trial

**DOI:** 10.3390/ijerph20032196

**Published:** 2023-01-26

**Authors:** Andrea Raisi, Tommaso Piva, Jonathan Myers, Valentina Zerbini, Simona Mandini, Tamara Zappaterra, Gianni Mazzoni, Elisabetta Tonet, Rita Pavasini, Gianluca Campo, Giovanni Grazzi, Emilio Paolo Visintin

**Affiliations:** 1Center for Exercise Science and Sport, University of Ferrara, 44123 Ferrara, Italy; 2Division of Cardiology, VA Palo Alto Health Care System, Palo Alto, CA 94304, USA; 3Department of Cardiovascular Medicine, Stanford University School of Medicine, Stanford, CA 94305, USA; 4Healthy Living for Pandemic Event Protection (HL-PIVOT) Network, University of Illinois at Chicago, Chicago, IL 60612, USA; 5Department of Humanities, University of Ferrara, 44121 Ferrara, Italy; 6Public Health Department, AUSL Ferrara, 44121 Ferrara, Italy; 7Cardiovascular Institute, Azienda Ospedaliero Universitaria di Ferrara, 44124 Ferrara, Italy

**Keywords:** cardiovascular disease, myocardial infarction, cardiac rehabilitation, qualitative research, adherence, lifestyle change

## Abstract

Traditional cardiac rehabilitation (CR) programs effectively improve physical performance and outcomes after myocardial infarction (MI). However, older patients are less likely to participate in such programs. The aim of this qualitative analysis was to investigate experiences and perceptions of cardiac outpatients enrolled in an innovative and exercise-based CR program and to identify possible barriers to improving adherence and quality of life. Semi-structured interviews were conducted on a sample of 31 patients (84% male; age 76 ± 6 years) from the Physical Activity Intervention in Elderly after Myocardial Infarction (PIpELINe) trial, after about six months of the event. Three main themes were identified: Personal feelings after the event; lifestyle change and perception of barriers; and relationships with familiars. Participants perceived sensations of fear at the time of their diagnosis and showed awareness of the importance of following specific health suggestions. They reported a significative change in previous habits and highlighted the need for periodic controls. Few of them felt insecure in carrying out daily activities or practicing exercise and reported an unnecessary protection from the family members. These findings will provide valuable insights for the development of a more feasible patient-centered CR model of intervention.

## 1. Introduction

Cardiovascular Disease (CVD) is the leading cause of mortality globally and represents a major health challenge, with a high prevalence of myocardial infarction (MI) among older people [1]. Indeed, in recent years, the mean age of patients admitted to the hospital for MI has significantly increased; more than half of patients with MI are aged ≥ 70 years [2]. Individuals with MI are more likely to report anxiety and depression and have a lower mental and physical quality of life [3]. They are also characterized by functional decline and reduced independence due to physiological ageing [4]. These factors can lead to the development of more sedentary behavior. For post-MI individuals, recovery of health is considered a struggle, including dealing with the management of daily problems [5]. Furthermore, these experiences make many patients uncertain and frightened of recurrence and death, whereas others are unaware that they are chronically ill [6].

Cardiac rehabilitation and secondary prevention (CR) applying a multifactorial approach [7] make important contributions to the continuum of care in post-MI patients and are a class IA recommendation [8]. However, despite compelling evidence about its health benefits, only 20–50% of individuals adhere to such recommendations [9]. Many patients achieve a modest lifestyle change, but they relapse into old habits when they start to feel better, failing the achievement of CR targets in the long term [10]. Therefore, as the benefits of CR are related to how closely a patient adheres to treatment, alternative programs that optimize adherence have been advocated [11,12]. One of the core components of these programs should be the focus on the individualized needs of a given patient. This is mainly based on the biopsychosocial model [13]. It directly contrasts the more traditional biomedical model that is characterized by a reductionist approach, in which illness is attributed to a single cause located within the body [14]. In the biopsychosocial model, health is instead seen as a scientific construct and a social phenomenon that involves a multidisciplinary approach, with the purpose of investigating both psychological and social factors that affect the patient [14,15].

As reported in the literature, many studies have investigated the perceptions and experiences of cardiac patients in relation to the management of symptoms and the stable modification of their lifestyle [16], or their engagement in CR programs [17]. Patients’ experiences of fear, uncertainty, and disconnection after the acute event have been described in most of the studies [16], as well as the common barriers that can limit a stable change and reduce the quality of life, such as psychological distress [18], lack of time due to work and other commitments, boredom with ongoing exercise regimes and other physical conditions such as arthritis or back problems [16]. As regards self-efficacy, for most patients achieving control of their own health is seen as a struggle, including negotiating the management of daily problems [19]. Finally, the feelings of independence and long-term change can be influenced by the behavior of the caregiver or family members, with episodes of excessive control and protection after the event which could undermine independence and self-commitment to long-term change [20,21]. Literature has therefore emphasized the importance of developing person-centered long-term programs, along with more tailored assessment and communication, in order to enhance adherence and lifestyle change. However, several studies focused on a sample of younger participants (aged less than 65 years old) [20,21,22], while others considered patients in an early stage after the acute event (within 6-8 weeks from the event) [20,23], or people involved in traditional CR interventions [24].

In order to address typical drawbacks of traditional CR, the “Physical Activity Intervention for Elderly Patients with Reduced Physical Performance after Acute Coronary Syndrome (HULK)” study has been conducted. Assessments of the efficacy of the HULK study have demonstrated that an early and feasible intervention based on a small number of supervised sessions, combined with an individualized home-based physical activity program, effectively improved older patients’ physical performance and quality of life [25,26,27,28]. More recently, this program has been updated in the “Physical Activity Intervention in Elderly after Myocardial Infarction (PIpELINe)” trial, which considers a tailored and low-cost multi-domain intervention, combined with exercise training.

Thus, to further support the quantitative research, the current qualitative study was designed. The purpose was to analyze feelings and experiences of older MI outpatients enrolled in an innovative and exercise-based CR program and to identify barriers or limitations that may reduce adherence and quality of life. Indeed, a qualitative study allows us to gain in-depth knowledge about the experiences, perceptions, and emotions of older MI outpatients. Specifically, we investigated patients’ interpretations of their illness within their recovery path, their capacity to manage daily problems, and their relationship with family members. Therefore, this study will provide a novel contribution by investigating the psychological and emotional experiences of older MI outpatients enrolled in the program and emphasizing the importance of a person-centered approach.

## 2. Materials and Methods

### 2.1. Study Population

This qualitative analysis was conducted on a sample of patients from the intervention group from the PIpELINe study, an ongoing multicenter, randomized clinical trial. A detailed description of the study has been reported elsewhere (www.clinicaltrials.gov (23 January 2023); NCT04183465). The purpose of the trial is to investigate if an early, tailored, and low-cost multi-domain intervention combining physical exercise, nutritional counseling, and risk factors control reduces cardiovascular death and rehospitalization for cardiovascular causes in older MI patients as compared to standard of care. Inclusion criteria were: (1) Age ≥ 65 years; (2) myocardial infarction (ST-segment elevation vs. no ST-segment elevation, STE or NSTE-MI) with indication to invasive management and (3) Short Physical Performance Battery (SPPB) between 4 and 9 at 1-month after hospital discharge. Exclusion criteria were: (1) Multivessel coronary disease with an indication for surgical revascularization or staged percutaneous coronary intervention (PCI); (2) life expectancy < 1 year; (3) severe aortic or mitral valvopathy; (4) left ventricular ejection fraction (LVEF) < 30%; (5) chronic heart failure NYHA III-IV; (6) severe cognitive impairment (Short Portable Mental Status questionnaire score < 4); and (7) inability to perform physical activity due to physical impairment. Patients in the intervention group received clinical and dietary counseling and home-based exercise prescription. Following the screening visit, all participants were referred to the Center of Sport and Exercise Science, University of Ferrara, Italy. The exercise intervention provides six supervised physical activity sessions, respectively, at 30, 60, 90, 180, 270, and 360 days after the hospital discharge. During each visit, patients perform a moderate and perceptually regulated 1 km Treadmill-Walk Test (1k-TWT). The test is validated for indirect exercise capacity estimation, calculated through an equation considering the distance, age, height, weight, and heart rate [29]. Patients unable to complete the test at a walking speed ≥ 3.0 km/h can perform the test over 500 m or 200 m [30,31,32]. Based on the results of the test, patients are encouraged to replicate similar walking sessions at home. In addition, a selection of calisthenics exercises based on the Otago Exercise Program [33] is performed, with the recommendation to be replicated at home, to improve strength and balance. The program is regularly adjusted during subsequent follow-up visits. Self-reported weekly leisure-time physical activity (wLTPA) is assessed from the 7-day physical activity recall questionnaire [34]. During each visit, patients receive an educational intervention with the purpose of reinforcing the idea of being physically active. Written informed consent was obtained from all patients at the time of enrollment. All data collected were treated confidentially and anonymized.

### 2.2. Interview Procedures

A semi-structured interview composed of six questions was developed. Questions were defined according to the biopsychosocial model, considering the importance of assessing feelings and experiences of the patient as a person, toward a more inclusive and efficient intervention. In more detail, they aimed to address: (i) Patients’ perceptions about their condition, its causes, and treatment; (ii) impressions about the intervention program, and about physical or psychological obstacles to a stable lifestyle change; and (iii) experiences about the relationship with caregivers, focusing on support and possible overprotection. An interview guide is described in Table 1.

Interviews were administered after approximately six months following the event to allow patients to manage the new “scenery” in which they were familiarized with the program. Data were collected from 31 outpatients between September 2021 and June 2022 during a routine follow-up visit at the Center of Sport and Exercise Science. We aimed to reach a substantial sample size of about 25–30 patients, in line with the literature and qualitative analysis guidelines [35,36]. A doctoral candidate (A.R.), with clinical and research experience with cardiac patients and actively collaborating to the PIpELINe intervention, conducted all the interviews individually. The interviewer was trained to conduct interviews by E.P.V. who has expertise in psychological qualitative research. Specifically, the interviewer was trained to be open to discussion, flexible in switching the order of questions or in providing cues to interviewees if needed, and sensitive to interviewees’ needs, emotions, and willingness to talk. As the interviewer is a collaborator in the PIpELINe trial, he already knew the interviewees before conducting the interviews. Participation in interviews was completely voluntary. Interviews were audio recorded and patients were free to interrupt at any time in case of discomfort.

### 2.3. Data Analysis

All recordings were transcribed verbatim. Transcripts were independently reviewed by A.R. and E.P.V. to ensure the accuracy and consistency of each interview. Interviews were read and re-read by A.R., and data were subsequently coded and grouped by their thematic similarity in order to describe the content of each interview. Finally, the main themes and subthemes were defined. After a first assessment, they were revised by S.M. who helped refine themes and supported data analysis. As the interviews were conducted and transcribed in Italian, all the extracts reported in the article have been converted to English by a translator, in order to maintain their precise meaning.

## 3. Results

### 3.1. Descriptive Statistics

Thirty-one outpatients with diagnoses of MI (74.6% of the outpatients in the intervention group when interviews were conducted) took part in the interviews. Characteristics of the participants are described in Table 2. Almost all the patients in the study had undergone percutaneous coronary intervention with drug-eluting stents (93.5%), while a small percentage (12.9%) reported a previous diagnosis of myocardial infarction.

### 3.2. Qualitative Findings

Three themes were developed: (1) Personal feelings after the event; (2) lifestyle change and perception of physical or psychological barriers; and (3) relationships with persons familiar with the participant: Support and overprotection. Themes are schematized in Figure 1 and described below with representative participant quotes.

### 3.3. Main Theme 1: Personal Feelings after the Event

When asked to talk about feelings and emotions after the diagnosis of the cardiovascular event, the answers given by patients highlighted similar reactions in the perception of what happened. Most of them (*n* = 19) described a sensation of fear, displeasure, and even astonishment at the time of the diagnosis.


*P3 (male, 78 years old): The truth? A bit of a scare, so that when I still think about it, I feel like crying. This was my feeling, fear and the urge to cry.*



*P5 (female, 88 years old): Bewilderment, disbelief, a bolt from the blue. But my nature is to react. I said to myself “I’ll do what needs to be done” and the procedure (operation and hospitalization) began.*


Some of the participants (*n* = 12) expressed calm and a sense of safety, especially those who were already hospitalized for MI or underwent cardiovascular operations.


*P23 (female, 79 years old): Overall, I have to say that I was surprised but I kept pretty quiet…When they told me that I had to stay there they said they would have to warn my husband, but I told them to go slowly and not to aggravate the situation…the experience was quite calm, I didn’t get agitated.*



*P10 (male, 73 years old): No feelings, it was a normal thing to me…I felt no emotions at all after the diagnosis, because I was so used to it like the first time I was operated on, delighted with all the operators, doctors and nurses, when I had the stroke. I was there as if I were home.*


### 3.4. Main Theme 2: Lifestyle Change and Perception of Physical or Psychological Barriers

This is the core part of the interview, as the project aimed to achieve a stable modification of the lifestyle and maintenance of healthy habits as well as the capacity to overcome physical or psychological obstacles.

Health suggestions and sharing of thoughts. Almost all participants (*n* = 27) showed awareness of the importance of following specific indications given by cardiologists at the time of the discharge. Such indications were mainly related to eating habits, risk factor control and physical activity. No participant had difficulties talking with family members about this.


*P1 (male, 72 years old): They suggested to change my diet slightly: less fat and more vegetables; to stop smoking as well, although I still smoke 4–5 a day, and to lead a dynamic life: moving around and walking without pushing myself…I then quietly talked about it with my family when they came to visit.*



*P7 (male, 77 years old): I focused on nutrition. I changed it radically. Everything that was heavy like sausages or cheese, I eliminated them. I don’t take them anymore, and my wife said we must continue like this. The cardiologist wrote it down for me and I’m putting it into practice. I quietly talked about it with my wife and daughter. I also noticed improvements in weight size: I can walk easier, I can do the movements, I can breathe better, all of these things here.*


Change in motivation, perception of barriers. Participants who initially were not motivated to modify their lifestyle (*n* = 16), described a change in previous habits, but also highlighted various obstacles. Such complications were especially related to fear or physical fatigue and difficulty in carrying out normal daily activities.


*P17 (male, 83 years old): I would say yes, because I’m not the type to cry on, I try to move and react. I try to leave it behind and move on…I’ve been perceiving physical obstacles, especially in my legs during walks or when I go dancing.*



*P25 (female, 75 years old): I’d rather have died, because I used to do a lot of things in the countryside, now I can’t anymore, and I get nervous. Not doing the things I used to do hurts me.*


Desire of periodic controls, efficacy of the program. The majority of patients (*n* = 23) perceived the need for periodic controls, in order to reduce the risk of recurrence and increase their feeling of safety. Furthermore, almost all (*n* = 26) stated that the program is feasible and effective and their willingness to attend visits was high.


*P5 (female, 88 years old): Ah yes! It is reassuring, because you never know. Luckily there are these checks and so I feel protected.*



*P1 (male, 72 years old): I think that control visits are valid because they encourage the patient. […] I continue to improve slowly, and this is important. Because we know that these are things that will never go away, but If you keep improving it means that the program works.*



*P13 (female, 76 years old): Initially, I didn’t feel comfortable with it, so much so that the first time I failed. Then I gradually changed my mind. So, I have a positive view.*


Self-efficacy. For patients, achieving control of their own health is perceived as a struggle, including dealing with common limitations of older age such as frailty and cognitive impairment. Despite this, most patients (*n* = 27) stated that they feel confident in carrying out daily activities, and few interviewees (*n* = 3) were afraid to engage in physical activity.


*P6 (male, 76 years old): I feel very confident, but I don’t have the chance. My legs collapse beneath me. As for the heart I feel confident for a simple reason, the doctor who operated on me said I am lucky, that I have a strong heart. As regards to physical activity, I don’t really like it, because I would rather rest, but I still do it.*



*P27 (male, 79 years old): I feel absolutely confident doing everything, to see that step by step I am recovering, and I am coming back, not as much as before but I feel I can withstand greater efforts.*


### 3.5. Main Theme 3: Relationship with Familiars: Support and Overprotection

Overprotection was a consistent observation and surveillance, and, in some cases, it contributed to unpleasant feelings and a consequent reduction in quality of life. Some of our patients (*n* = 7) in fact described a higher than necessary protection, especially from their partner or children.


*P2 (male, 79 years old): Obviously I feel a lot of apprehension when others (family members) tell me “Be careful doing this, be careful doing that”. These things bother you a little bit, I understand that they do it for my own good, but it also hurts me. Overall, I feel constrained by my family, sometimes I start off with enthusiasm to do something and then I am told “but be careful, don’t do this, don’t do that”.*


However, most participants (*n* = 24) did not report excessive pressure, and in some cases (*n* = 2), they perceived higher apprehension as a positive factor.


*P9 (male, 79 years old): When I was feeling well my children never asked me how I was, now they ask me how my breathing or my heart is. I don’t perceive too much protection, but on the other hand there is more apprehension on their part. For me it is a positive sign.*


## 4. Discussion

The purpose of this qualitative analysis was to investigate in-depth the experiences of outpatients enrolled in the CR intervention program, by analyzing their capacity to change their lifestyle, modify their previous habits, and attempt to overcome typical limitations that affect their peers. As previously described, most of the qualitative analyses in the literature are focused on younger patients or involved in traditional center-based CR [20,21,22,23,24]. We aimed to extend previous research by analyzing the perspective of older patients, who often have unique needs and more numerous and daunting barriers to participation in any intervention [37], within an innovative home-based CR program.

In terms of feelings after the acute event, the current results are consistent with other studies [20,21]. These studies describe how the perception of an acute cardiovascular event is often interpreted as worse than it is, as several cardiac patients held a belief that a diagnosis of myocardial infarction resulted in instant death [20]. Our patients reported a homogeneous response to this theme, including a general sensation of fear and bewilderment, but also trust in the hospital staff. Concerning the lifestyle change and the perception of barriers, the responses suggest that nearly all the participants were confident in terms of their prognosis and efficacy of the treatment. They made some changes, especially regarding nutrition and exercise, after the advice received by cardiologists and they had no problems sharing thoughts and concerns with friends and family members.

Previous literature demonstrates that changing lifestyle is challenging and difficult [38]. Behavioral change can be hard to perform as psychological distress from life baggage, along with social and practical barriers, can influence the ability to change [18]. In addition, adherence, defined as a complex behavioral process, is strongly influenced by the environment in which patients live, including healthcare providers practice, and how healthcare systems deliver care [39]. For this, a successful program has to keep the focus on the needs of the patient and on the collaboration between different providers. Given that many traditional CR programs are considered to be too expensive or too strenuous and long-term adherence is often poor, the key factor appears to be the design of the program. Most of our participants were in fact more likely to attend visits, and improvements were often a positive element in their rehabilitation path. Another fundamental point is the analysis of self-efficacy, defined as the capacity to exercise control over events that affect patients’ lives [40]. Self-efficacy is a well-studied concept within cardiac rehabilitation, and evidence from several studies indicates a positive association between perceived self-efficacy and personal goals during CR programs, such as increased amount of physical activity [41], better food choices [42], reduced smoking habits [43], and improved quality of life [44]. Almost all the patients in our sample affirmed feeling confident in carrying out daily activities, along with maintaining their social life and improving their physical activity level. This demonstrates the efficacy of continuous motivational counselling that allowed us to reinforce the importance of continued involvement in both in physical and social activities.

Finally, it is important to consider the relationship with a patients’ partner and family. This is a very important step in the recovery of the patient. Support offered by family members is often a key point for the achievement of long-term adherence and improved lifestyle [45]. Unfortunately, previous studies suggest that many patients refer feelings of frustration for being overprotected by their partners/families in the early recovery period [20], and this is consistent with our results. Therefore, in order to increase family support and to decision sharing, patients were encouraged to bring their partner and/or a significant family member to each visit and they were free to ask questions or express doubts, or anxieties related to the rehabilitation process.

### 4.1. Strengths of the Study

The novelty of this study is the collection of qualitative data within a clinical trial based on the development of an early and tailored CR program in MI patients. Participants focused on their past and on personal experiences, which allowed the investigator to gain a deeper understanding of their history not only from a clinical perspective but also from a psychological and humanistic point of view. These results should therefore be compared with quantitative data, in order to analyze the efficacy of the motivational approach on the improvement of clinical outcomes.

### 4.2. Limitations of the Study

This analysis has some limitations which should be acknowledged. First, the interviews were performed by only considering the intervention group, as the control group did not attend the exercise-based program and it was therefore not possible to contact outpatients from the control arm. Second, older patients with cognitive decline, disability or severe left ventricular dysfunction were excluded, due to the difficulty to attend visits and perform functional evaluations. Third, there was a relatively small proportion of women in the study. This is relevant because previous studies suggest that female patients have lower age-standardized cardiovascular disease incidence, prevalence, and death rates than men [46], but overall they receive less care, and they are less able to maintain lasting lifestyle changes [47,48]. Fourth, while the interviewer encouraged participants to express their feelings and experiences as freely as possible in a non-judgmental way, social desirability might still play a role, and account for responses suggesting high levels of hope, control, and self-efficacy. Overall, these considerations limit the generalizability of our findings.

## 5. Conclusions

Understanding patients’ experiences and interpretation of their illness may be essential in order to improve MI rehabilitation and secondary prevention programs. These results reinforce the effectiveness of this approach and provide valuable insights for the development of a more feasible patient-centered model of intervention.

## Figures and Tables

**Figure 1 ijerph-20-02196-f001:**
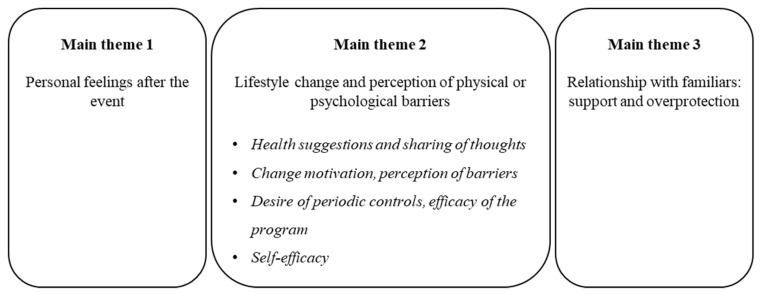
Main themes and sub-themes identified in qualitative interviews.

**Table 1 ijerph-20-02196-t001:** Semi-structured interview guide.

Which sensations did you feel immediately after the diagnosis of the cardiac event?
2. After the operation and the hospitalization, which health-related suggestions did you receive and from whom? Did you talk to anyone about it?
3. Did the event generate in you a motivation to lifestyle changing? After 6 months, did you perceive any obstacles? If so, which ones?
4. After the event, did you perceive the desire to be periodically checked? If so, what are your impressions about the program that you are joining?
5. After the event, how confident do you feel about carrying out your usual daily activities? And what about physical activity?
6. After the event, how do you consider your relationship and how do you feel treated by your family members and those who take care of you? Do you perceive any changes?

**Table 2 ijerph-20-02196-t002:** Characteristics of patients at the time of the interview. Values are presented as mean (SD) or percentage. ARB, angiotensin receptor blocker; BMI, body mass index; PTCA, percutaneous transluminal coronary angioplasty, stenting, or both.

Variable	All Participants (*n* = 31)
** *Demographics* **	
Age (yr)	76 (6)
Gender (M/F)	26/5
BMI (kg/m^2^)	26.6 (4.7)
Instruction (yr)	9 (4)
Marital status (married, %)	80.6
Current working (%)	6.5
** *Risk factors* **	
Family history (%)	12.9
Hypertension (%)	90.3
Diabetes (%)	48.4
Current smoking (%)	29.0
** *Medical history* **	
PTCA (%)	93.5
Previous infarction (%)	12.9
** *Medications* **	
ACE inhibitor or ARB (%)	90.3
Aspirin (%)	71.0
β-blocker (%)	74.2
Calcium antagonist (%)	29.0
Statin (%)	96.8
Diuretic (%)	25.8

## Data Availability

Data collected during the current study will be available from the corresponding author on reasonable request.

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
