# Peer review of "Experience and Perceptions among Older Outpatients after Myocardial Infarction following an Exercise Intervention: A Qualitative Analysis from the PIpELINe Trial"

_ijerph, 2023, doi:10.3390/ijerph20032196_

Round 1

Reviewer 1 Report

In this paper, the authors performed a semi-structured interview to examine the experiences and perceptions of 31 cardiac outpatients and identify possible barriers to improving adherence and quality of life. The reviewer agrees with the authors that older patients with MI are less likely to participate in outpatient cardiac rehabilitation programs due to many barriers. This manuscript provides an important contribution to the rapidly growing field of older cardiac rehabilitation patients, but the authors need to clarify several concerns regarding this study.

As background to the planning of this study, it is unclear how much is known and how much is newly verified in this study. Therefore, it would be better to add in the background what this study is newly trying to reveal.

Data analysis methods for semi-structured interviews are unclear.

For example, the manuscript mentions "identified main themes" how did you identify the main themes? Moreover, there are various methods, from transcripts review to data analysis. For example, did the authors do any processing after creating verbatim transcripts from the raw speech data obtained? This is important to ensure the reproducibility of the study.

The interviewers were part of the doctoral project, but did they practice the interviewing process?

Semi-structured interviews require the interviewer's judgment, experience, and tact to proceed flexibly.

This study is being conducted on the intervention group of an ongoing multicenter, randomized clinical trial. There are some serious concerns associated with this point.

1. Was this interview conducted with all patients in the intervention group? If it does not interview all patients in the intervention group, it may be necessary to 1) present a flowchart of the subjects and 2)clarify what percentage of the intervention group agreed to participate in the study. In addition, those who agreed to participate in the semi-structured interview will receive a separate exposure, the semi-structured interview, six months after treatment, in addition to the main RCT. Thus, the intervention group patients who enrolled in this study had different exposure to the semi-structured interview than the intervention group without the semi-structured interview. Is this a problem in terms of the study design of the RCT?

2. Why was the non-intervention group not interviewed? There may have been differences in the perception of barriers depending on the presence or absence of outpatient interventions.

3. Was the content of this semi-structured interview directly related to the intervention content of the intervention group?

The interview guide in Table 2 should be presented in a method section.

Discussion

The authors state that the novelty is that they collected qualitative data from clinical trials, but they do not present the results of the crucial clinical trials. If this novelty is to be claimed, it is essential to consider the results in the context of clinical trials. If the results of clinical trials cannot be presented, another novelty should be clearly stated.

Reviewer 2 Report

In methods section of both abstract & body of text consider including the study design.

1. What is the main question addressed by the research?---Manuscript focused on perceptions of older adults following an exercise intervention.

2. Do you consider the topic original or relevant in the field?---Yes topic is relevant

Does it address a specific gap in the field?---the importance of exercise following an acute cardiac event is important. Improvements in the type of exercise and engagement of participants is addressed in this manuscript.

3. What does it add to the subject area compared with other published material?---adds to the current body of knowledge

4. What specific improvements should the authors consider regarding the methodology? In methods section of both abstract & body of text consider including the study design.

What further controls should be considered?---N/A

5. Are the conclusions consistent with the evidence and arguments presented and do they address the main question posed?---Yes

6. Are the references appropriate?---Yes

7. Please include any additional comments on the tables and figures.---no additional comments.

Reviewer 3 Report

Abstract

1.       Withdraw Secondary prevention, I would talk about this in the introduction, that Cr is secondary prevention and I would just use the term CR.

Introduction:

1.       Just say that Cr is SP, but use the term CR.

2.       I missed a good justification for doing a qualitative study.

3.       I would talk about Biopsychosocial model in methods.

Methods

1.       Who conducted the interview? Where was it held? Did the interviewer know the interviewees? Was it done individually or in a group? If in a group, how many people? All this information is missing from the method.

Discussion1.  You point out a research proposal different from the objectives. review both
